# Democratizing Usage of Planning Systems by Facilitating Research in Algorithm Selection for Planning (Discussion Topic)

**Michael Katz**
IBM Research
Yorktown Heights, NY, USA
michael.katz1@ibm.om

**Silvan Sievers**
University of Basel
Basel, Switzerland
silvan.sievers@unibas.ch

Planning research has produced a large number of tools for various formalisms. As planning is a computationally challenging task, it is important to come up with a variety of ideas and approaches to tackle the various sources of planning tasks' complexity. However, as a by-product, it is unclear even to an experienced planning researcher what tool will work well on a new planning task. The challenge is even harder for a layman. Most planning tools are not easily accessible and those that are might have inadequate performance on some tasks. The problem was partially addressed by the most recent International Planning Competition, with the competing planners being made publicly available in Singularity containers, allowing for easily building and running the planners. This, however, does not solve the problem of choosing the right planner for a given task. An impatient user might forgo the option of using domain-independent planners altogether as a result of an inadequate performance of one randomly chosen planner.

Online algorithm selection using machine learning techniques was shown to be able to produce planners that show good performance on previously unseen domains. For optimal planning, previous success stories were mostly exploiting the fact that most tasks, if solved, are solved quickly. Thus, being able to accurately predict planner performance opens new perspectives for better exploiting various existing planners in practice. The research in this field does, however, currently require a rather deep familiarity with the field of planning.

In order to allow researchers from outside of the planning community to tackle the problem of planner performance prediction, we should alleviate the dependence on special knowledge in planning. One step towards achieving this goal is to provide data consumable by machine learning tools and complying with their assumptions made. The data consists of data points that each represents a planning task. Labels of each data point could represent performance of some planner on that task and features of the task, to help the user determining which features allow which planners to perform well on the task.

The most important assumption about data in machine learning is independent and identical distribution. Such an assumption is unrealistic when planning domains are created manually. Another assumption is that the data is representative of the entire population. In domain-independent planning, where the population consists of all tasks representable in the language of choice (e.g., PDDL), creating planning domains manually cannot produce representable data. For both assumptions to be satisfied, it is required to produce data automatically, and in a way that will cover a variety of possible planning tasks.

We propose a new track at IPC to help achieving the mentioned goals. The track will provide an easy access to existing planners, as well as to the data – a variety of hand-crafted, as well as automatically generated planning tasks, with additional information on the performance of these planners on the existing tasks. If providing this information seems to be unrealistic (e.g., due to computational load, as all planners would need to be run under the same conditions), an alternative could be to provide the instructions of how to obtain the performance information by running the planners (provided as containers as in the most recent IPC). Furthermore, instead of only providing a fixed set of benchmarks, one could also provide generators to automatically generate more data.

Participants in this track would submit domain-independent planners, possibly building on the provided planners, which would be evaluated on new domains like in the classical track of previous IPCs. In some sense, this new track would be similar to the learning track of previous IPCs, but with planners being provided alongside benchmarks, and with the goal of creating domain-independent rather than domain-dependent planners. The uniqueness of this track should be in alleviating the need for special knowledge in planning. The goal is to both achieve better exposure and to ease the use of planning tools outside of the planning community.