# OpenReview forum: "Democratizing Usage of Planning Systems by Facilitating Research in Algorithm Selection for Planning"
_icaps-conference.org/ICAPS/2019/Workshop/WIPC_

### Official Review · AnonReviewer2 · 2019-04-25
**Democratizing Usage of Planning Systems by Facilitating Research in Algorithm Selection for Planning**

**Rating:** 7
**Confidence:** 3

**Review:**


This abstract considers the important and challenging problem of enabling non-expert users to select what will be a good domain-independent planning system to use on a given new domain. The proposal is to have planners, accompanied by domains/problem instances and data points, that allow users to identify which planners will then be good on another domain.  I think it will make for interesting discussion at the workshop.  A couple of points for discussion:

1) It's not clear to me what exactly the format of the proposed data would be: a domain/instance and the performance on that domain is still not necessarily that helpful to someone new to planning: how does the new user link this to their own new domain?  Is there somehow a plan to extract features from the domain so that the user can do similar and then determine which is good for their domain?  Will this necessarily be easier for a user than just running several planners?

2) I'd like to understand more about how this becomes a competition track as that wasn't clear from the abstract: do people submit planners, domains, data points?  Or systems that take these and predict performance on new unseen domains?  What might a metric look like for winning this competition?

3) Is such a track something the organisers are willing to get behind running (or setting up a system so that this can be automated)?  Only it can be hard to find volunteers to do this!

Overall, I think this could make for interesting discussion and the problem is certainly an important one, but it would be nice to see the abstract extended a little before the workshop to cover a few more details (e.g. answering questions 1 and 2 above) in order to make it clearer what is being proposed.

---

### Official Review · AnonReviewer3 · 2019-04-25
**My review needs a title?**

**Rating:** 7
**Confidence:** 5

**Review:**

This abstract proposes an interesting idea that would be useful in the workshop.

I actually think the idea of automatically generating planning tasks is really important. There are domain structures that receive less focus only because they're not IPCs, or because our current planners aren't very good at them; yet are pertinent to potential applications of planning.

A few typos:

Large amount of tools -> Large number of tools
planning tasks complexity -> planning tasks' complexity
to experienced planning researcher -> to an experienced planning researcher